# *Coelastrella terrestris* for Adonixanthin Production: Physiological Characterization and Evaluation of Secondary Carotenoid Productivity

**DOI:** 10.3390/md20030175

**Published:** 2022-02-26

**Authors:** Philipp Doppler, Ricarda Kriechbaum, Maria Käfer, Julian Kopp, Daniel Remias, Oliver Spadiut

**Affiliations:** 1Research Division Biochemical Engineering, Institute of Chemical, Environmental and Bioscience Engineering, TU Wien, Gumpendorfer Strasse 1a, 1060 Vienna, Austria; philipp.doppler@tuwien.ac.at (P.D.); ricarda.kriechbaum@tuwien.ac.at (R.K.); maria.kaefer@students.boku.ac.at (M.K.); julian.kopp@tuwien.ac.at (J.K.); 2School of Engineering, University of Applied Sciences Upper Austria, Stelzhamerstr. 23, 4600 Wels, Austria; daniel.remias@fh-wels.at

**Keywords:** microalgae, stirred photobioreactor, nutrient starvation, osmotic stress, unsaturated fatty acids, astaxanthin, canthaxanthin

## Abstract

A novel strain of *Coelastrella terrestris* (Chlorophyta) was collected from red mucilage in a glacier foreland in Iceland. Its morphology showed characteristic single, ellipsoidal cells with apical wart-like wall thickenings. Physiological characterization revealed the presence of the rare keto-carotenoid adonixanthin, as well as high levels of unsaturated fatty acids of up to 85%. Initial screening experiments with different carbon sources for accelerated mixotrophic biomass growth were done. Consequently, a scale up to 1.25 L stirred photobioreactor cultivations yielded a maximum of 1.96 mg·L^−1^ adonixanthin in free and esterified forms. It could be shown that supplementing acetate to the medium increased the volumetric productivity after entering the nitrogen limitation phase compared to autotrophic control cultures. This study describes a promising way of biotechnological adonixanthin production using *Coelastrella terrestris*.

## 1. Introduction

Microalgae belong to the oldest forms of life on earth and are genetically highly diverse [1]. They are the largest group of microorganisms capable of photosynthesis, which enables them to harvest light energy for fixation of inorganic carbon dioxide (CO_2_) for biomass growth [2]. For the production of microalgal biomass, many different photobioreactor (PBR) designs have been developed [3]. Additionally, different strategies of autotrophic, mixotrophic and heterotrophic cultivation regimes were introduced for enhanced biomass production [4,5]. For mixotrophic cultures, pentoses, hexoses, or other substrates were reported to boost growth of numerous microalgal species [6,7].

The natural product portfolio of microalgae is extremely versatile. Whole biomass can be used as a human nutraceutical food source since it is rich in vitamins, essential amino acids and polyunsaturated fatty acids (FA) [8]. It can be processed for intended production of next-generation biofuels, or individual bioactive compounds for pharmaceutical applications can be extracted [9,10]. An abundant group of these compounds are pigments, namely chlorophyll (Chl) and carotenoids [11]. Carotenoids are polyunsaturated tetraterpenoids with about 850 different derivatives [12]. The carotenoids produced by microalgae consist of carotenes and xanthophylls. The primary carotenoids (PC) of the chloroplasts, such as β-carotene, lutein or zeaxanthin, are essential for photosynthesis and have industrial relevance in cosmetics and pharmaceutics [13,14]. 

Secondary carotenoids (SC) can accumulate in certain cell compartments, e.g., cytosolic lipid bodies or in plastoglobuli after stress induction [15,16]. The most important and effective stress stimuli are excess of light, nutrient starvation, temperature and pH shifts or osmotic pressure [17,18]. The most well-known SC is astaxanthin, a natural pigment used as colorant in food production, in cosmetics or as feed additive for aquaculture [19]. Astaxanthin has the highest antioxidant capacity of all carotenoids [20]. It is produced during aplanospore formation of the prominent green alga *Haematococcus lacustris*, commonly known as *Haematococcus pluvialis*. Another microalgal SC with high potential in industry is adonixanthin [12]. It is a unique keto-carotenoid containing two polar ionone rings. Compared to the astaxanthin structure, only one of the ionone rings carries a ketone. This causes polarity and could result in stronger protective effects than astaxanthin [21]. Adonixanthin was reported to show very promising results in anti-tumor and anti-cancer treatment and protection against brain damage induced by cerebral hemorrhage [21,22]. However, studies on the biotechnological production of adonixanthin in algae are scarce.

In this study a novel strain of the green alga *Coelastrella terrestris* (Chlorophyta, Scenedesmaceae) was characterized. Strain WP154.1 was sampled in Iceland and isolated from red mucilage in a small brook close to a glacier. The species was first described by Reisigl (1964) in the Austrian Alps and initially named *Scotiella terrestris* [23]. In recent years, several members of the genus *Coelastrella* were screened for pigment and lipid production. Hu et al. (2013) identified the pigment composition of the thermotolerant strain *Coelastrella* sp. F50 grown in glass bottles [24]; Minhas et al. (2016) identified five *Coelastrella* sp. isolates and conducted Erlenmeyer flask cultures for pigment and lipid measurements [25]; Wang et al. (2019) did phylogenetic studies and described a new species of the genus *Coelastrella* [26]; Goecke et al. (2020) published the FA pattern and qualitative pigment composition of their strain grown in a 250 mL tubular PBR [27]; and Zaytseva et al. (2021) reported on their new strain *C. rubescens* NAMSU R1 and identified the carotenoids present after Erlenmeyer flask cultivations [28]. However, the only report on adonixanthin accumulation of *Coelastrella* was published by Minyuk et al. (2017) [29]. The group did *C. rubescens* Erlenmeyer flask experiments with inorganic CO_2_ or organic acetate as a carbon source (C-source) in combination with low-pH stress for SC production. However, they did not find any correlation of C-source and SC productivity, and mainly focused on the identification of the accumulated SC and lipids rather than the potential of biotechnological production. 

Our goal was to characterize strain *C. terrestris* WP154.1 in terms of morphology and physiology, pigment content and fatty acid composition. The potential of this novel strain as alternative source of natural pigments, especially adonixanthin, was investigated for future biotechnological production. Thus, we performed screening experiments to find adequate C-sources for enhanced mixotrophic biomass formation compared to autotrophic growth. Subsequently, this knowledge was used for scaling up to 1.25 L PBR experiments. This study describes a biotechnological adonixanthin production process by *C. terrestris*. 

## 2. Results

### 2.1. Morphological and Physiological Characterization of C. terrestris WP154.1

#### 2.1.1. Morphology

Light microscopical observations showed the unicellular alga in its characteristic shape (Figure 1). Single cells of *C. terrestris* WP154.1 appeared ellipsoidal or lemon-shaped, both apices showed wart-like wall thickenings. Their cell size ranged between 9 to 13 µm length and 7 to 10 µm width (Figure 1a). Maturing cells enlarged in size underwent asexual reproduction by autosporulation (Figure 1b,c). Figure 1d shows the division into four daughter cells in planar arrangement, but also eight or more daughter cells were observed. By cell wall rupture, the daughter cells with a chloroplast and a prominent pyrenoid were released (Figure 1e). Due to experimental nutrient starvation, the aged cells accumulated SC and appeared in orange-reddish color (Figure 1f).

#### 2.1.2. Pigment Composition

To identify the molecules which were responsible for WP154.1’s coloration, orange colored biomass was treated for pigment extraction and identification. The chromatogram after HPLC separation of the extract is shown in Figure 2.

The detected pigments were Chl *a* and Chl *b*, the PC included neoxanthin, violaxanthin, lutein, zeaxanthin and β-carotene, and the SC comprised astaxanthin, adonixanthin, canthaxanthin and echinenone (Table 1). Astaxanthin and adonixanthin were also detected in esterified forms. Monoesters (ME) were found for both and diesters (DE) only for astaxanthin.

#### 2.1.3. Fatty Acid Profile of Lipids

For the quantification of the FA composition of *C. terrestris*, a representative sample of orange-colored biomass harvested at the end of a cultivation was analyzed. The measured values of the corresponding FA methyl esters (FAME) are shown in Figure 3. All saturated even-numbered FA between C14 and C22 were detected with different levels of unsaturation. 

In total, 234.9 mg∙g^−1^ FAME were measured, of which 127.1 mg·g^−1^ (54.1%) were polyunsaturated (PUFA), 73.7 mg·g^−1^ (31.4%) monounsaturated (MUFA), and only 34.2 mg·g^−1^ (14.5%) of all FA were fully saturated (Figure 3a). The most abundant FA was the MUFA oleic acid (C18:1) with 70.3 mg·g^−1^, which reflects 30.0% of all FAMEs (Figure 3b). The second-most quantified FAME was the PUFA α-linoleic acid (C18:3) with 57.8 mg·g^−1^, followed by 35.8 mg·g^−1^ linoleic acid (C18:2). 

### 2.2. Carbon-Source Screening for Mixotrophic Growth

Erlenmeyer flask experiments were done to determine suitable C-sources with the major focus to find accelerated mixotrophic biomass growth compared to an autotrophic reference process. The investigated C-sources were the hexoses fructose and glucose, the pentoses ribose and xylose, as well as acetate and glycerol. Growth curves expressed by optical density values at 600 nm (OD_600_) for each of the substrates including an autotrophic control culture were tracked for 37 d to identify the substrate for fastest mixotrophic growth (Figure 4). For comparison, the initial exponential growth rates (µ_max_) between the start of the experiment and day 10 were calculated. This time range was chosen, since all cultures still contained nitrate and supplemented C-source, and thus growth was not limited by nutrient starvation (data not shown).

The culture supplemented with glucose was growing fastest and resulted in a µ_max,glucose_ = 0.34 d^−1^, followed by fructose (Table 2). The autotrophic control showed higher growth rates than cultures provided with ribose, acetate, glycerol and xylose. Biomass was increasing in all cultures for the first 21 d, with glucose and fructose showing roughly the double OD_600_ values compared to the autotrophic control. Quantification of the C-source concentrations of glucose, fructose and acetate revealed values below the limit of detection after 21 d of cultivation. For ribose, xylose and glycerol, only approximately 10% of initial carbon content was metabolized after 21 d (data not shown). Thus, subsequent mixotrophic cultivations were conducted with (i) glucose, (ii) acetate, which is a common C-source in algae cultivations, and (iii) an autotrophic control.

### 2.3. Scale-Up to 1.25 L Lab-Scale Photobioreactors

To scale up the cultivations of *C. terrestris*, externally illuminated stirred PBRs were used. The experiments were conducted at two different nitrate levels of Bold’s Basal Medium (BBM) with standard BBM and BBM with doubled nitrate content (2N-BBM). Both media were supplemented with glucose or acetate as mixotrophic C-sources. For comparison, autotrophic controls were done resulting in a total of six controlled and scalable PBR cultivations. The algal biomass trends expressed as OD_600_ for these cultivations are shown in Figure 5.

All cultures showed exponential growth from the beginning. During the first days of cultivation, the maximum growth rates (µ_max_) for BBM and 2N-BBM were highest in cultures containing glucose. The average µ_max_ was 0.72 d^−1^ and 0.76 d^−1^, respectively. These were closely followed by the autotrophic controls with 0.63 d^−1^ and 0.68 d^−1^ for BBM and 2N-BBM, respectively. Both bioreactors containing acetate showed reduced µ_max_ of 0.56 d^−1^ and 0.55 d^−1^. After a process time of 7 d, the nitrate was completely consumed in all cultivations, which led to reduced growth rates (Table 3). From this time on, SC accumulated in the cells. OD_600_ values in this phase were still increasing to maximum OD_600_ values of about 7.5 for BBM and 10.0 for 2N-BBM. Acetate in the medium reduced the final OD_600_ of the cultures to approximately 4.3 and 6.1, respectively. 

Extracted pigments were individually quantified and subsequently grouped in the three categories Chl, PC and SC (Figure 6 and Appendix A). PBR cultivations with 2N-BBM gave more total pigments than with BBM (Figure 6a). The highest pigment-producing culture was the 2N-BBM autotrophic control with 31.34 mg·L^−1^ of total pigments concentration. The lowest-producing run of all experiments was BBM supplemented with acetate, with only 8.42 mg·L^−1^. The average pigment composition for Chl, PC and SC was approx. 61 ± 5%, 14 ± 3% and 25 ± 5%, respectively. After having a closer look on the volumetric SC content (Figure 6b), the 2N-BBM acetate run showed highest overall values with 6.39 mg·L^−1^, which was slightly more than the control (6.14 mg·L^−1^). 2N-BBM runs had more volumetric SC content with the exception of BBM glucose, which was overall third highest with 5.53 mg·L^−1^ SC. Lowest SC accumulating culture was BBM containing acetate (1.99 mg·L^−1^). Concerning adonixanthin, the maximum concentration of 1.96 mg·L^−1^ was detected in 2N-BBM with acetate, followed by the 2N-BBM control (1.64 mg·L^−1^). Relative ratios of the SC astaxanthin, adonixanthin, canthaxanthin and echinenone of all cultivations were calculated as 46 ± 5%, 28 ± 2%, 23 ± 5% and 2 ± 2%.

For evaluating the productivities of all SC and in particular adonixanthin, the corresponding volumetric (r) and specific production rates (q) were calculated (Table 3). After 7 d, nitrate was consumed and the SC accumulation phase started until final harvest after 21 d. This resulted in a maximum value of r_SC_ for the 2N-BBM acetate culture with 0.43 mg·L^−1^·d^−1^. For adonixanthin, the highest volumetric (r_adonixanthin_) and specific production rates (q_adonixanthin_) observed were 0.13 mg·L^−1^·d^−1^ and 0.06 mg·g^−1^·d^−1^ in the 2N-BBM acetate culture after stress induction.

## 3. Discussion

### 3.1. Morphological and Physiological Characterization

The green algal strain WP154.1 was identified as *Coelastrella terrestris* (Chlorophyta, Scenedesmaceae) by molecular means. Light microscopical examinations confirmed that cytomorphological characteristics were in agreement with previous descriptions [23,30]. The extracted pigments of orange-colored biomass revealed the presence of the rare SC adonixanthin. After lipid extraction and quantification, more than 54% of FA were polyunsaturated, with less than 15% saturated. These values were also comparable to other reports of members of the genus *Coelastrella* [25]. 

This initial physiological characterization indicates great potential for future industrial use of *C. terrestris* WP154.1 for both, scalable adonixanthin and polyunsaturated FA production. Due to its climate, Iceland seems to be a promising origin of microbial strains useful for biotechnology, as recently shown for another green microalga [31].

### 3.2. Prelimary Experiments and Scale up to Photobioreactors

Axenic cultivation experiments started with a screening for suitable C-sources regarding accelerated mixotrophic biomass growth compared to autotrophic controls. Mixotrophy could solve the bottleneck for a future economically feasible process [32]. The C-sources fructose, glucose, ribose, xylose, acetate and glycerol were supplemented in Erlenmeyer flask experiments. 

The hexoses glucose and fructose performed best concerning the growth rate µ_max_, which was 48% and 34% higher than the autotrophic control. All other C-sources caused reduced growth in relation to the control. Acetate, a commonly used C-source for algae, was lowering µ_max_ by 40%. Ribose, xylose and glycerol were not metabolized by the cells, despite reports showing consumption by other green algal species [6,33,34]. Therefore, subsequent experiments were done with either (i) glucose addition for highest biomass growth rates, or (ii) acetate supply as the most common C-source for carotenoid production, and (iii) an autotrophic control.

The PBR cultivation processes were designed to autoinduce SC production after nitrate in the medium was consumed. Growth rates in PBR were essentially higher compared to Erlenmeyer flask experiments. Comparing the BBM autotrophic cultures in both scales, the growth rate of the PBR cultivation was raised by a factor of 2.7. This was reasoned to be due to the improved light availability and enhanced mass transfer of gaseous CO_2_ in the used stirred PBR [3,35]. In the PBR, highest growth rates were observed in mixotrophic cultivations with glucose. However, biomass content of the autotrophic control was not significantly lower during the nitrogen-limited phase. Acetate as a mixotrophic C-source reduced growth rates and final biomass concentration for both nitrate levels. Apparently, acetate caused additional stress in the form of osmotic pressure, which impaired algal growth [36]. After pigment quantification, the autotrophic 2N-BBM culture revealed the highest overall volumetric pigment content primarily caused by elevated Chl levels. This was due to lowered Chl production of microalgae during mixotrophic conditions in the other PBR experiments [37]. During the nitrogen starvation phase, Chl was degraded and SC accumulated, as reported for green algae capable of SC synthesis [38]. Although acetate did not show the highest values for biomass concentration, specific SC content was considerably increased, and thus volumetric SC amount was highest of all runs with 6.39 mg·L^−1^ total SC and 1.96 mg·L^−1^ (31%) adonixanthin. Adonixanthin was approx. a factor of 11 higher compared to a recently published study of the green alga *T. minimum*, accumulating adonixanthin to a maximum volumetric content of 0.179 mg·L^−1^ [36]. After entering the nitrogen starvation phase, the volumetric SC and adonixanthin production rates in the acetate culture were boosted by a factor of 9.0 to 0.13 mg·L^−1^·d^−1^ and a factor of 6.9 to 0.06 mg·L^−1^·d^−1^, respectively. This increase was much higher compared to glucose and the autotrophic control and again ascribed to addition of acetate in the culture which applied osmotic stress. Furthermore, this explains why acetate is often the stress factor of choice, e.g., for SC production processes for astaxanthin [39]. Additionally, it is cheap and often a byproduct of industry, e.g., during plant biomass hydrolyzation [40]. This might be crucial for economic and sustainable production processes in the future. Surprisingly, Minyuk et al. reported that the closely related *Coelastrella rubescens* was not affected by osmotic stress caused by acetate [29].

### 3.3. Biotechnological Potential of C. terrestris WP154.1

Microalgal biomass or extracted components are regarded as high-value products for human health and nutrition [41]. Specific compounds show bioactivity and are already used as functional food or as pharmaceutics [42,43]. *C. terrestris* WP154.1 revealed high content of the rare keto-carotenoid adonixanthin which has reportedly great potential to treat cancer or protect against brain damage [21,44]. To further improve adonixanthin production rates, we recommend two-stage cultivations to apply osmotic stress right after the growth phase ends [45]. Another possibility for optimization could be triggering the cells to transform to the resting cell stage, similar to *Haematococcus lacustris*; however, this cell stage has not been reported for any *Coelastrella* species yet [39,46].

## 4. Materials and Methods

### 4.1. Sampling, Strain Isolation and Cultivation Medium

The microalga was collected in Iceland in the foreland of Sölheimjökull glacier from a small brook where macroscopic mucilaginous mats of reddish cyanobacteria dominated. The GPS position was N63°31.890 W19°22.081 at approximately 120 m above sea level. The permission for collection in Iceland was granted by the National Energy Authority Orkustofnun, leyfisnúmer OS-2017_L020-01. Cells were isolated from field material by autotrophic cultivation at 15 °C and 30–40 µmol PAR·m^−2^·s^−1^ (14 h light·d^−1^) on 1.6% agar plates with Bold’s Basal Medium (BBM), which contained 250 mg·L^−1^ NaNO_3_, 75 mg·L^−1^ MgSO_4_·7H_2_O, 25 mg·L^−1^ NaCl, 75 mg·L^−1^ K_2_HPO_4_, 175 mg·L^−1^ KH_2_PO_4_, 25 mg·L^−1^ CaCl_2_·2H_2_O, and 1 mL·L^−1^ of 1000× trace metal mix with a composition of 8.82 g·L^−1^ ZnSO_4_·7H_2_O, 1.44 g·L^−1^ MnCl_2_·4H_2_O, 0.71 g·L^−1^ MoO_3_, 1.57 g·L^−1^ CuSO_4_·5H_2_O, 0.49 g·L^−1^ Co(NO_3_)_2_·6H_2_O, 11.4 g·L^−1^ H_3_BO_3_, 50.0 g·L^−1^ Na_2_EDTA·2H_2_O, and 4.98 g·L^−1^ FeSO_4_·7H_2_O [47]. 

The unialgal culture WP154.2 was established by mechanical isolation of cell colonies with a sterile loop. Axenic cultures were established by a workflow utilizing fluorescence-activated cell sorting combined with plate spreading [48].

### 4.2. Strain Identification

The strain identification of *C. terrestris* WP154.2 (accession number OM574907.1) was done by molecular means with an 18S rDNA marker. The primers used for this were 18F2 (5’-AAC CTG GTT GAT CCT GCC AGT-3’) and 18R2 (5’-TGA TCC TTC TGC AGG TTC ACC TAC G-3’), as described by [49]. The sequenced fragment with 1,067 bp length was submitted to the National Center for Biotechnology Information (NCBI). The closest blast hits in the database of NCBI were *Coelastrella terrestris* strain CCALA 476 (accession JX513882.1) and *Coelastrella terrestris* strain KZ-5-4-9 (accession number MK231276.1). Both hits showed 100% identity at a query cover of 99%. The resulting phylogenetic tree was created by the NCBI tool via the neighbor joining method [50]. It shows calculated distances of related species in Appendix A. The strain WP154.2 is clearly grouping within the genus *Coelastrella*. Its closest neighbors were *Coelastrella terrestris* strain CCALA 476 (accession JX513882.1) and *Coelastrella terrestris* strain KZ-5-4-9 (accession number MK231276.1).

### 4.3. Light Microscopy

For light microscopical observations, a CKX41 inverted microscope (Olympus, Tokyo, Japan) with 1000x total magnification, was equipped with an EOS 250D camera (Canon, Tokyo, Japan). The camera was operated via EOS utility software (Canon, Tokyo, Japan).

### 4.4. Pigment Extraction, Identification and Quantification

The whole procedure of pigment extraction, identification and quantification is described in detail elsewhere [36]. Briefly, total pigments were extracted in acetone by disrupting roughly 20 mg lyophilized biomass between glass beads in a FastPrep-24 instrument (MP Biomedicals, Santa Ana, CA, USA). The acetonic extract was separated on a 1290 Infinity II LC System (Agilent Technologies, Santa Clara, CA, USA) equipped with an Acclaim C30, 3 µm, 2.1 mm × 100 mm column (Thermo Fisher Scientific, Waltham, MA, USA). The masses were detected by a 6545 LC/Q-TOF mass spectrometer (Agilent Technologies, Santa Clara, CA, USA) operated in atmospheric pressure chemical ionization (APCI) positive mode. The pigments were separately identified by comparison of detected and theoretical masses in combination with characteristic spectral absorption of each substance in agreement to literature [51]. Carotenoid mono- and diesters were assigned to adonixanthin and astaxanthin according to the characteristic absorption spectra of the respective unesterified form [52]. Astaxanthin was quantified on a Vanquish Flex HPLC system with DAD at 450 nm (Thermo Fisher Scientific, Waltham, MA, USA) with standard calibration. All other carotenoids were reference to astaxanthin at 450 nm [36]. Chlorophyll *a* and *b* were analyzed photometrically [53]. 

### 4.5. Lipid Extraction, Transesterification and Quantification of Fatty Acids

The workflow for quantifying fatty acids as methyl esters was done according to [36]. About 50 mg dried biomass was extracted for lipids with a chloroform-methanol solvent. After transesterification in a methanol-hydrochloric acid mixture, the sample was injected to a 7890A/5975C GC-MS system (Agilent Technologies, Santa Clara, CA, USA) equipped with a CTC Combi PAL autosampler. Identification was done by a reference library and quantification via flame ionization detector. Calibration was achieved by the standard substances methyl palmitate (C16:0), methyl stearate (C18:0) and methyl arachidate (C20:0) (Sigma-Aldrich, St. Louis, MO, USA). Three independent biomass samples were treated for GC-MS quantification.

### 4.6. C-Source Screening

The experiments were conducted in 250 mL Erlenmeyer flasks. Every flask contained 50 mL standard BBM with 250 mg·L^−1^ NaNO_3_ at pH 7.0. Additional C-sources fructose, glucose, ribose, xylose, acetate and glycerol were added to the final concentration of 1.0 g·L^−1^ before autoclaving. After sterilization, the vitamins were added via 0.22 µm syringe filter. All flasks were inoculated with 2 mL preculture grown in BBM in the middle of the exponential growth phase with OD_600_ about 0.5. The flasks were placed in a Minitron incubation shaker (Infors, Basel, Switzerland) at 20 °C, 150 rpm (25 mm amplitude) with 3% CO_2_-enriched atmosphere. The provided illumination expressed as photon flux density (PPFD) was 30 µmol PAR·m^−2^·s^−1^ for 14 h light·d^−1^. Samples were taken after 10 and 21 d. The biomass content was estimated in 1 mL cuvettes on a Nanodrop One photometer (Thermo Fisher Scientific, Waltham, MA, USA) by absorption measurements at 600 nm wavelength. Then, 1 mL aliquots were centrifuged, and the supernatant was stored separately for subsequent C-source and nitrate concentration measurements. Every cultivation experiment was carried out in biological duplicates and the respective average values were used for data evaluation.

### 4.7. Stirred Photobioreactor Cultivations

For cultivations in stirred PBR, a Ralf multi-bioreactor system (Bioengineering, Wald, Switzerland) was used (Appendix A). The glass vessel had 2.0 L total volume with an inner diameter of 95 mm. Light was provided from the outside through the glass jacket for heating and cooling. A 5 m warm-white LED light strip (Paulmann, Völksen, Germany) providing 960 lm via 100 single LED spots, was uniformly wrapped around the vessel. The reactor was equipped with two 55 mm turbine impellers, an EasyFerm plus pH probe (Hamilton, Bonaduz, Switzerland) and a ring sparger for gas supply. Gas was premixed by two separate type 4850 mass flow controllers connected to a 0254 operation unit (Brooks Instruments, Hatfield, PA, USA).

Standard BBM, or BBM with double nitrate content (2N-BBM), was directly used or supplemented with 33.3 mmol·L^−1^ carbon in form of glucose (C_6_H_12_O_6_) or acetate (CH_3_COONa). Then, a 1.25 L prepared medium was filled into the PBR prior to autoclaving. After that, vitamins were injected aseptically via 0.22 µm syringe filter and a septum. The gas was mixed to 3.0% CO_2_-enriched air and provided at a rate of 0.1 vvm. Cultivation temperate was 20 °C and the stirrer agitated with 300 rpm. The provided PPFD in the center of the reactor was about 170 µmol PAR·m^−2^·s^−1^ during the first 72 h of the process and then changed to 850 µmol PAR·m^−2^·s^−1^, measured by an US-SQS/L sensor (Walz, Ulm, Germany) in a cell-free medium. The light was automatically simulating a 14 h light·d^-1^ cycle. The pH was set to 6.3 ± 0.05 via automatic addition of 1 mol·L^−1^ Na_2_CO_3_ prior to inoculation. Roughly 20 mL preculture, also grown in BBM, was added via a syringe and septum. The final OD_600_ after inoculation was between 0.02 and 0.04. The experiment was started when the respective culture reached OD_600_ of 0.10 ± 0.01 to avoid possible preculture differences. Regular samples for OD_600_ (1 mL cuvettes on a Nanodrop One photometer (Thermo Fisher Scientific, Waltham, MA, USA)) and nitrate quantification were drawn on days 0, 1, 2, 3, 4, 5, 6, 7, 9, 11, 13, 15 and 21. Then, 20 mL biomass samples for pigment quantification were harvested by centrifugation and lyophilized on days 7, 11, 15 and 21. Specific exponential growth rate (*µ*_max_) was calculated with OD_600_ at a certain process time (OD_600,*t*_) and compared to the initial value (OD_600,*t*=0_) (Equation (1)):*µ*_max_ = ln(OD_600,*t*_/OD_600,*t*=0_)/*t*(1)

The average volumetric production rate of a product (*r*_P_) between two time points (*t*_1_ and *t*_2_) was calculated after product concentration measurements of the product of interest at the same time points (*c*_P,1_ and *c*_P,2_):*r*_P_ = (*c*_P,2_ − *c*_P,1_)/(*t*_2_ − *t*_1_)(2)
and for specific production rate (*q*_P_), the biomass (*X*) was considered additionally (Equation (3)):*q*_P_ = *r*_P_/*X*(3)

### 4.8. Quantification of Carbon-Source

The supernatant was analyzed on an UltiMate 3000 HPLC system with diode array detector (DAD) (Thermo Fisher Scientific, Waltham, MA, USA) by an Aminex HPX-87 H column (Bio-Rad Laboratories, Hercules, CA, USA). The method was recently described in detail [54]. Calibration standards of all investigated C-sources (glucose, fructose, ribose, xylose, acetate and glycerol) were freshly prepared of pure substances (Sigma-Aldrich, St. Louis, MO, USA), diluted and analyzed accordingly.

### 4.9. Nitrate Quantification

The nitrate content in the cultures were determined in the supernatant after centrifugation of the samples. The exact method was published elsewhere [36]. A Dionex ICS-6000 ion chromatography system (Thermo Fisher Scientific, Waltham, MA, USA) equipped with an IonPac AS11 and guard column (Thermo Fisher Scientific, Waltham, MA, USA) was used. Detection was done via a conductivity detector unit (Thermo Fisher Scientific, Waltham, MA, USA). Quantification was possible via calibration of nitrate standards (Sigma-Aldrich, St. Louis, MO, USA). 

## Figures and Tables

**Figure 1 marinedrugs-20-00175-f001:**
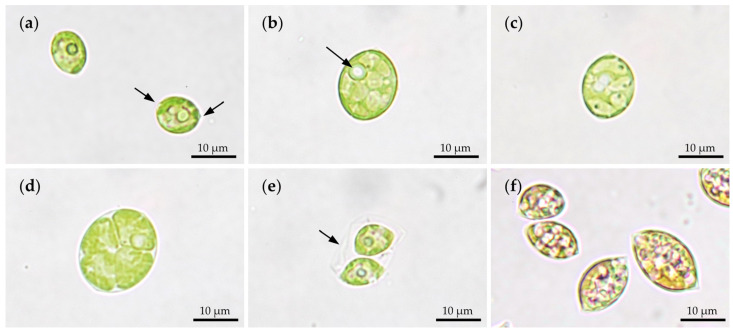
Light microscopical observation of *C. terrestris* WP154.1 at different cell stages. (**a**) Single lemon-shaped cells with definite chloroplast and pyrenoid, arrows: wart-like wall thickenings; (**b**,**c**) swollen cells before cell division, arrow: pyrenoid; (**d**) four daughter cells after cell division in planar arrangement; (**e**) two daughter cells within old cell wall of mother cell, arrow: cell wall of mother cell; (**f**) stressed cells with accumulated secondary carotenoids, indicated by their orange coloration.

**Figure 2 marinedrugs-20-00175-f002:**
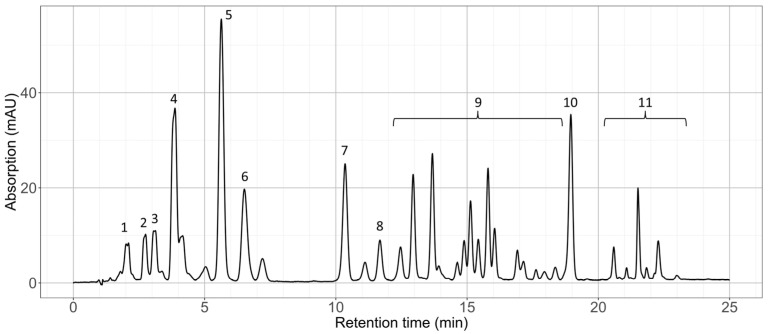
Absorption chromatogram at 450 nm of chromatographic separation before mass spectrometric analysis of the total pigment extract of *C. terrestris* WP154.1.

**Figure 3 marinedrugs-20-00175-f003:**
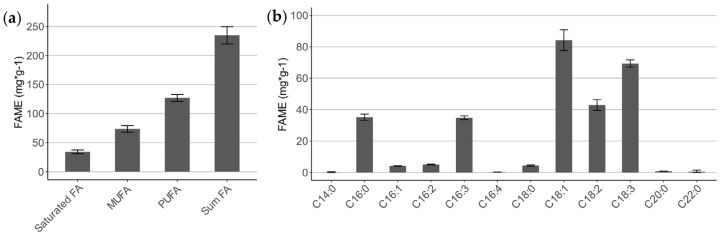
Fatty acid (FA) analysis of total lipids of *C. terrestris* as methyl esters. (**a**) Quantification of FA classified in degree of unsaturation; (**b**) quantification of all detected FA variants. MUFA, monounsaturated FA; PUFA, polyunsaturated FA.

**Figure 4 marinedrugs-20-00175-f004:**
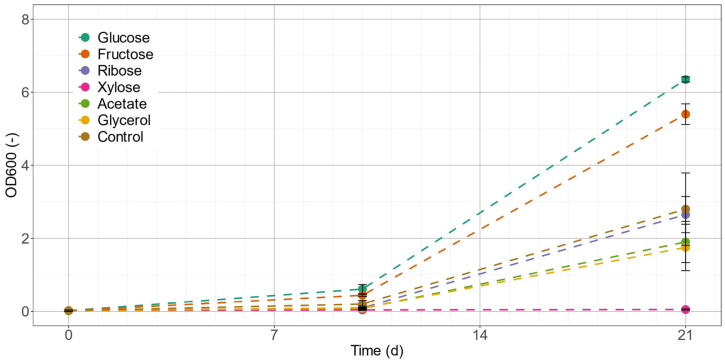
Growth curves of *C. terrestris* in Erlenmeyer flask cultures during initial growth screening with different C-sources. The biomass content of the culture is expressed as measure of optical density at 600 nm (OD_600_).

**Figure 5 marinedrugs-20-00175-f005:**
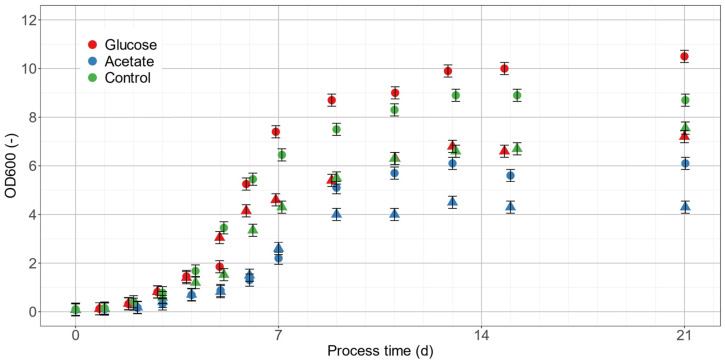
*C. terrestris* growth curves of PBR cultivations. Cultivations in BBM (filled triangles) and in 2N-BBM (fill circles) with supplemented C-sources glucose (red), acetate (blue), or control (green).

**Figure 6 marinedrugs-20-00175-f006:**
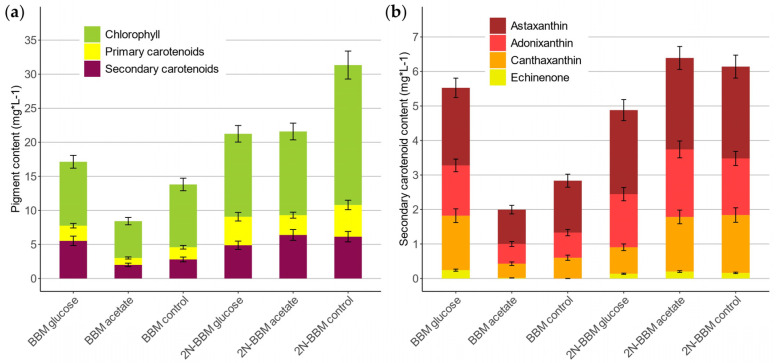
Quantification of pigment content after PBR cultivations of *C. terrestris*. (**a**) Total volumetric content of the pigments Chl, PC, SC; (**b**) volumetric content of SC astaxanthin, adonixanthin, canthaxanthin and echinenone.

**Table 1 marinedrugs-20-00175-t001:** Identified pigments of acetonic extract of *C. terrestris* WP154.1 by HPLC-MS analysis.

	Assigned Substance	Detected Mass (M + H)^+^	Theoretical Mass (M + H)^+^	Mass Error (ppm)
1	Neoxanthin/violaxanthin ^1^	601.4242	601.4251	1.5
2	Astaxanthin	597.3930	597.3938	1.3
3	Adonixanthin	583.4343	583.4351	1.4
4	Lutein/zeaxanthin ^1^	569.4355	569.4353	0.4
5	Canthaxanthin	565.4046	565.4040	1.1
6	Chlorophyll *b*	907.5218	907.5219	0.1
7	Chlorophyll *a*	893.5424	893.5426	0.2
8	Echinenone	551.4238	551.4247	1.6
9	Astaxanthin and adonixanthin ME			
10	β-carotene	537.4441	537.4455	2.6
11	Astaxanthin DE			

^1^ co-eluting isomers.

**Table 2 marinedrugs-20-00175-t002:** Growth evaluation of *C. terrestris* C-source screening. The listed parameters are optical density (OD_600_) after 21 days of cultivation and exponential growth rate (µ_max_) during this period.

	Glucose	Fructose	Ribose	Xylose	Acetate	Glycerol	Control
OD_600_ after 21 d (−)	6.35	5.40	2.65	0.05	1.90	1.75	2.80
µ_max_ (d^−1^)	0.34	0.31	0.16	0.07	0.14	0.15	0.23

**Table 3 marinedrugs-20-00175-t003:** Growth and productivity assessment of *C. terrestris* WP154.1 PBR cultivations. The list shows exponential growth rate (µ_max_), average linear growth rate during limitation phase (µ_limitation_), volumetric production rate of SC (r_SC_) and adonixanthin (r_adonixanthin_), and specific production rate of SC (q_SC_) and adonixanthin (q_adonixanthin_). Growth phase was between 0 to 7 d and the limitation phase lasted from 7 until 21 d.

	Interval (d)	BBM			2N-BBM		
		Glucose	Acetate	Control	Glucose	Acetate	Control
µ_max_ (d^−1^)	0–7	0.717 ± 0.089	0.561 ± 0.070	0.626 ± 0.078	0.757 ± 0.094	0.554 ± 0.069	0.681 ± 0.085
µ_limitation_ (d^−1^)	7–21	0.111 ± 0.013	0.117 ± 0.014	0.126 ± 0.015	0.100 ± 0.012	0.197 ± 0.024	0.096 ± 0.012
r_SC_ (mg·L^−1^·d^−1^)	0–7	0.184 ± 0.023	0.042 ± 0.005	0.035 ± 0.004	0.178 ± 0.022	0.047 ± 0.005	0.127 ± 0.015
	7–21	0.302 ± 0.037	0.121 ± 0.015	0.182 ± 0.022	0.259 ± 0.032	0.432 ± 0.054	0.374 ± 0.046
q_SC_ (mg·g^−1^·d^−1^)	0–7	0.184 ± 0.023	0.070 ± 0.008	0.056 ± 0.007	0.242 ± 0.030	0.087 ± 0.010	0.137 ± 0.017
	7–21	0.123 ± 0.015	0.067 ± 0.008	0.072 ± 0.009	0.097 ± 0.012	0.188 ± 0.023	0.128 ± 0.016
r_adonixanthin_ (mg·L^−1^·d^−1^)	0–7	0.046 ± 0.005	0.011 ± 0.001	0.005 ± 0.000	0.049 ± 0.006	0.018 ± 0.002	0.056 ± 0.007
	7–21	0.080 ± 0.010	0.034 ± 0.004	0.049 ± 0.006	0.085 ± 0.010	0.130 ± 0.016	0.089 ± 0.011
q_adonixanthin_ (mg·g^−1^·d^−1^)	0–7	0.046 ± 0.005	0.019 ± 0.002	0.009 ± 0.001	0.067 ± 0.008	0.034 ± 0.004	0.060 ± 0.007
	7–21	0.033 ± 0.004	0.019 ± 0.002	0.019 ± 0.002	0.032 ± 0.004	0.056 ± 0.007	0.030 ± 0.003

## Data Availability

The data presented in this study are available on request from the corresponding author.

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
