# Peer review of "Coelastrella terrestris for Adonixanthin Production: Physiological Characterization and Evaluation of Secondary Carotenoid Productivity"

_marinedrugs, 2022, doi:10.3390/md20030175_

Round 1

Reviewer 1 Report

COELASTRELLA TERRESTRIS FOR ADONIXANTHIN PRODUCTION: PHYSIOLOGICAL CHARACTERIZATION AND EVALUATION OF SECONDARY CAROTENOID PRODUCTIVITY

The Authors isolated and characterized a new strain of the carotenogenic microalga Coelastrella terrestris (Scenedesmaceae) and characterized it in terms of carotenoid production and growth on different sources of organic carbon.

  1. It is the study of limited novelty suitable for a more specialized journal. Accumulation of the intermediates of astaxanthin bioynthesis in microalgae have been many times reported previously (https://doi.org/10.1023/A:1008173807143; https://doi.org/10.3390/biology10070643), especially in Coelastrella spp. (https://doi.org/10.1134/S1021443716040105; ) including most recent works (https://doi.org/10.1111/pre.12412; https://doi.org/10.3390/plants10122601). Unfortunately the Authors have provided recent background in the field neither in introduction nor in discussion. The idea of growth on different carbon sources (Na-acetate is also not novel. It was studied, e.g. in the work Minyuk et al. cited in the text.
  2. Another important issue is that methods do not correspond to state-of -the-art in the field: simple BLAST search is insufficient for identification of the strain. A phylogenetic analysis involving authentic strains is required. It has not been done. Moreover, the sequence must be submitted to NCBI GenBank. I could not find the GenBank ID in the text.
  3. There are more specific issues.

-I cannot understand whether the carotenoids were primary or secondary. It is particularly questionable for β-carotene which can be both secondary and primary.

-There are many specific drawbacks and mistakes.

-Title does not reflect the work: Authors studied carotenoids only, not whole repertoire of antioxidants of algae.

-l. 21-22. Antioxidant production from algae is a matter of several recent decades.

-l. 42. By definition ALL existed carotenoids are divided into carotenes and xanthophylls.

-l. 45. It is wrong. Particularly, Dunaliella accumulates β-carotene as a secondary carotenoid in plastoglobuli.

-l. 51, 265. Haematococcus lacustris is a wrong obsolete sinonym. Haematococcus lacustris is correct.

-l. 63. Some most recent works on Coelastrella are ignored.

-l. 97-103. Content of this subsection is not scientifically reported. It is only verbal description without table(s) or graphical presentation.

-l. 149-151. It is Methods.

-l. 221. Comma is missing.

-l. 222. The word Surprisingly does not sound scientific. It should be removed.

-"nitrogen in the medium". It is a jargonism. Nitrogen is a gas N2. "Inorganic nitrogen source in the medium" is correct.

-l. 254. Coelastrella rubescenS.

-l. 265. Actually haematocysts of H. lacustris are not cysts. They are aplanospores.

Author Response

The Authors isolated and characterized a new strain of the carotenogenic microalga Coelastrella terrestris (Scenedesmaceae) and characterized it in terms of carotenoid production and growth on different sources of organic carbon.

  1. It is the study of limited novelty suitable for a more specialized journal. Accumulation of the intermediates of astaxanthin bioynthesis in microalgae have been many times reported previously (https://doi.org/10.1023/A:1008173807143; https://doi.org/10.3390/biology10070643), especially in Coelastrella spp. (https://doi.org/10.1134/S1021443716040105; ) including most recent works (https://doi.org/10.1111/pre.12412; https://doi.org/10.3390/plants10122601). Unfortunately the Authors have provided recent background in the field neither in introduction nor in discussion. The idea of growth on different carbon sources (Na-acetate is also not novel. It was studied, e.g. in the work Minyuk et al. cited in the text.

The authors thank the Reviewer for this comment. Indeed, the detection of adonixanthin was already reported for microalgae. Adonixanthin is an intermediate of astaxanthin synthesis, as recently well explained by Maoka, T. (2020, https://doi.org/10.1007/s11418-019-01364-x). However, when an alga accumulates high values of astaxanthin, the intracellular concentrations of adonixanthin are relatively low (Grewe and Griehl, 2008, https://doi.org/10.1002/biot.200800067). In this study, our novel strain was accumulating considerable amounts of adonixanthin as well as astaxanthin (and canthaxanthin). This means adonixanthin was accumulated together with astaxanthin during secondary metabolism induced by nutrient starvation, which was rarely reported in literature before.

We were aware of the above listed literature regarding Coelastrella spp. at the time we submitted our manuscript. In the introduction we already have four publications of Coelastrella spp. in combination with carotenoid production cited. We added two of the three suggested publications in the intro. Kawasaki et al. (2019, https://doi.org/10.1111/pre.12412) was excluded since the authors only characterized a new species of Coelastrella but only did phylogenetic analysis with microscopic observations. No carotenoids were identified or quantified.

Additionally, our work differs from the work of Minyuk et al. (2017, cited in the text) significantly. They did uncontrolled Erlenmeyer flask experiments and detected only traces of adonixanthin. Moreover, they did not find any influence of carbon source on secondary carotenoid accumulation. The differentiation to state of the art is explained in the introduction.

  1. Another important issue is that methods do not correspond to state-of -the-art in the field: simple BLAST search is insufficient for identification of the strain. A phylogenetic analysis involving authentic strains is required. It has not been done. Moreover, the sequence must be submitted to NCBI GenBank. I could not find the GenBank ID in the text.

The authors appreciate this advice. We added phylogenetic analysis for strain identification to the manuscript and submitted the sequence to NCBI GenBank. The sequence was already processed and according to NCBI it will be available online from Feb 13th, 2022. The accession number can be found in the manuscript.

  1. There are more specific issues.

-I cannot understand whether the carotenoids were primary or secondary. It is particularly questionable for β-carotene which can be both secondary and primary.

During photobioreactor cultivations the production and degradation of carotenoids was tracked. The carotenoids which were produced by secondary metabolism induced by nutrient starvation (the stress phase of cultivation) are secondary carotenoids. In our case astaxanthin, adonixanthin, echinenone and canthaxanthin. Other pigments were produced by primary metabolism from the beginning.

-There are many specific drawbacks and mistakes.

-Title does not reflect the work: Authors studied carotenoids only, not whole repertoire of antioxidants of algae.

The chosen title does not mention “antioxidants”. We only focused on secondary carotenoids, especially adonixanthin.

-l. 21-22. Antioxidant production from algae is a matter of several recent decades.

No biotechnological production of adonixanthin exists. Therefore, research to find possible natural producers is of great interest. We rearranged the sentence for better understanding.

-l. 42. By definition ALL existed carotenoids are divided into carotenes and xanthophylls.

Revised.

-l. 45. It is wrong. Particularly, Dunaliella accumulates β-carotene as a secondary carotenoid in plastoglobuli.

Revised.

-l. 51, 265. Haematococcus lacustris is a wrong obsolete sinonym. Haematococcus lacustris is correct.

On www.algaebase.org both synonyms H. pluvialis and H. lacustris are accepted in the community. A PubMed search (on February, 9th 2022) showed for the last half decade 2017-2021 24 results for H. lacustris and 261 for H. pluvialis. Therefore, the wider accepted synonym H. pluvialis was used for a wider reach and impact of this manuscript.

-l. 63. Some most recent works on Coelastrella are ignored.

This comment was already addressed above (1st comment).

-l. 97-103. Content of this subsection is not scientifically reported. It is only verbal description without table(s) or graphical presentation.

A Table (Supplementary Table S2) and a Figure (Supplementary Figure S2 (was S1 in previous version)) are shown in the Supplementary Material. The verbal description is referring to this Table and Figure.

-l. 149-151. It is Methods.

Revised as requested.

-l. 221. Comma is missing.

Revised.

-l. 222. The word Surprisingly does not sound scientific. It should be removed.

Removed.

-"nitrogen in the medium". It is a jargonism. Nitrogen is a gas N2. "Inorganic nitrogen source in the medium" is correct.

Changed to “nitrate in the medium”.

-l. 254. Coelastrella rubescenS.

Corrected.

-l. 265. Actually haematocysts of H. lacustris are not cysts. They are aplanospores.

Corrected.

Reviewer 2 Report

Dear Authors,
your manuscript concerns a very stimulating research and the theme of natural pigments, and in particular of carotenoids, is current. 
You have conducted a very convincing research project and have reported very inspiring results. The production of adonixanthin from Coelastrella terrestris could be an interesting advance for biotechnologies.

In my opinion the manuscript could be accepted for publication but I suggest very few changes to the text (see attached file). Edit some references; some of them are incomplete (some authors are lacking) and in others the names of the algal species need to be italicized.

Author Response

Dear Authors,

your manuscript concerns a very stimulating research and the theme of natural pigments, and in particular of carotenoids, is current.

You have conducted a very convincing research project and have reported very inspiring results. The production of adonixanthin from Coelastrella terrestris could be an interesting advance for biotechnologies.

In my opinion the manuscript could be accepted for publication but I suggest very few changes to the text (see attached file). Edit some references; some of them are incomplete (some authors are lacking) and in others the names of the algal species need to be italicized

The authors thank the reviewer for her/his positive thoughts on the impact and importance of our manuscript! Comments and suggestions in the attached pdf-file were all revised as requested.

Reviewer 3 Report

Marinedrugs-1574177 is a well written manuscript. However, there are a few issues that prevent me from accepting the manuscript as is:

  1. The classification of the microalga as Coelastrella terrestris should be improved by showing a phylogenetic analysis using two different inference methods (ML and Bayesian-based inference are recommended). BLAST results are not enough to attribute the taxon a microalga belongs to, even if it is 99%, because one must exclude that the phylogenetic signal is strong enough to differentiate it from other Coelastrella microalgae using only 18S sequences.
  2. Please improve the inset legend of Fig. 1 by showing exactly where the pyrenoid is located.
  3. In Figs. 3 and 4, OD measurements were used while growing microalgae. The authors used a wavelength set at 600nm. However, this is not a good practice as OD may be measuring other light scattering objects that are not microalgae, like bacteria, and can be influenced by pigments that absorb light at this wavelength. Were the cultures monoaxenic without bacterial contamination? Moreover, I would like for the authors to present the data in dry weight if possible, as that is the "gold" standard, because OD measurement can be quite untrustworthy if it is not properly calibrated with dry weight data.
  4. On page 5, please explain what the BBM acronym is, because that's only explained on page 8 in M&M. Please on the latter page the composition of this growth medium. Do not just state its reference.
  5. Fig. 4 is quite odd. The legend refers to two panels and only one panel is shown. Moreover, the inset legend refers to BBM and 2N-BBM but no data points are visible for these media. I think a panel is missing. Please correct this. Moreover, the authors should be more careful in the way they present their axis legends. Use capitalized words and with a larger, more readable font.
  6. In Table 2, standard deviation should be shown, and a statistical analysis should be made in order to see which differences are statistically different.
  7. As monoaxenicity is quite important for mixotrophic growth to avoid bacterial contamination which could affect the measured growth, authors should explain in M&M how they isolated this microalga and how they ensured its monoaxenicity.

Author Response

Marinedrugs-1574177 is a well written manuscript. However, there are a few issues that prevent me from accepting the manuscript as is:

  1. The classification of the microalga as Coelastrella terrestris should be improved by showing a phylogenetic analysis using two different inference methods (ML and Bayesian-based inference are recommended). BLAST results are not enough to attribute the taxon a microalga belongs to, even if it is 99%, because one must exclude that the phylogenetic signal is strong enough to differentiate it from other Coelastrella microalgae using only 18S sequences.

The authors appreciate this suggestion. We added phylogenetic analysis for strain identification to the manuscript by maximum likelihood (ML) method.

  1. Please improve the inset legend of Fig. 1 by showing exactly where the pyrenoid is located.

We are grateful for this input. Figure 1 was changed accordingly. An arrow is now pointing towards the clearly visible pyrenoid in the swollen cell in Fig.1b.

  1. In Figs. 3 and 4, OD measurements were used while growing microalgae. The authors used a wavelength set at 600nm. However, this is not a good practice as OD may be measuring other light scattering objects that are not microalgae, like bacteria, and can be influenced by pigments that absorb light at this wavelength. Were the cultures monoaxenic without bacterial contamination? Moreover, I would like for the authors to present the data in dry weight if possible, as that is the "gold" standard, because OD measurement can be quite untrustworthy if it is not properly calibrated with dry weight data.

The authors are grateful for this input. We are aware of possible influences of OD600 measurements. We are always working with filtered water and axenic, this means, we only measure the absorption of our algal culture. OD600 gives fast and relatively exact values. Dry cell weight (DCW) determination via gravimetry is very error-prone and needs high sample volumes, especially for doing replicates. This is simply impossible for Erlenmeyer flask experiments. Additionally, we are not calculating DCW out of OD600, else we weighed the dried biomass before lipid and pigment extraction and therefore omitted this “calibration error”.

  1. On page 5, please explain what the BBM acronym is, because that's only explained on page 8 in M&M. Please on the latter page the composition of this growth medium. Do not just state its reference.

We added the full name of BBM on that page. Additionally, the composition of BBM is now specified in the Materials & Methods section 4.1., as requested.

  1. 4 is quite odd. The legend refers to two panels and only one panel is shown. Moreover, the inset legend refers to BBM and 2N-BBM but no data points are visible for these media. I think a panel is missing. Please correct this. Moreover, the authors should be more careful in the way they present their axis legends. Use capitalized words and with a larger, more readable font.

The authors thank for this comment. Fig. 4 and its caption were updated. Moreover, the axis legends were adapted with capitalized words and larger fonts (Fig. 3-5).

  1. In Table 2, standard deviation should be shown, and a statistical analysis should be made in order to see which differences are statistically different.

The authors acknowledge this input. Table 2 was extended and now shows the standard deviation of the different calculated rates.

  1. As monoaxenicity is quite important for mixotrophic growth to avoid bacterial contamination which could affect the measured growth, authors should explain in M&M how they isolated this microalga and how they ensured its monoaxenicity.

The isolation of this specific strain was already described in the Materials & Methods section in subsection 4.1. Sampling, Strain Isolation and Cultivation Medium. We always work with unialgal, axenic cultures. Indeed, cultivations wouldn’t be possible with glucose in the medium.

Round 2

Reviewer 1 Report

It has been decided to revise the article. More detailed analysis of presented data is necessary. Moreover, I cannot agree with some responses of the Authors.

-Although the Authors tried to explain the novelty of the work and cited some recent papers, the novelty is weak. (1) previously I provided some work on carotenoid composition in Coelastrella spp., reporting one more strain has only a local significance, (2) probably, the novelty could be related to use a system including the strain and photobioreactor; in this case they should provide its construction as a scheme or figure, not only verbal description, (3) adonixanthin accumulation is not novel for microalgae, particularly in significant amounts (see e. g. the work Yuan et al. (2002). Food Chemistry, 76(3), 319-325). Moreover, the Authors did mention even their own recently published paper (Doppler et al. (2021)., first reported member of hydrodictyaceae to accumulate secondary carotenoids. Life, 11(2), 107.) on Tetrahedron minutum, in which the fraction of adonixanthin was also significant. To highlight the novelty, they should at least compare obtained data with this publication. What was the difference (except one more strain of algae)? Unfortunately, the Authors cited this work in the methodological section only. Another possible novelty is the growth on different carbon sources, but they did not focus on it as could be seen from the title and abstract.

-First of all, I cannot understand the general organization of data. Microscopic photographs of the strain are located in the main body of the text, whereas more precise identification by phylogenetic analysis is given as supplementary. The same is true about the data of carotenoid identification by HPLC-MS (or LC-MS, I cannot understand from the text). Since carotenoid composition is discussed, this data plays a key role in the work, but chromatogram and results of MS are also given only as supplementary data. Moreover, these two parts of the work have to be substantially revised.

-Phylogenetic analysis: to determine the strain at the level of species, the phylogenetic tree should not include the sequences mentioned as Coelastrella sp., particularly closely related to the studied strain. The tree must contain authentic strains, these strains have to be indicated. CCALA 476 is not authentic. If it will not be done, the strain should be defined as Coelastrella sp.

-The procedure of phylogenetic analysis should be revised. "It was calculated by the Maximum Likelihood method by 1000 bootstrap replications" - the ML tree cannot be calculated by bootstrap procedure. It is built based on the calculation and maximisation of the LogLikelihood function for a selected DNA evolution model, whereas bootstrap is the procedure for testing the robustness of tree topology. The reference for the ML algorithm, bootstrap and selected model are required. What was the algorithm of multiple alignment, what was the criterion for model selection, how was the initial tree for ML analysis selected, what was the heuristic search method?

-Results of pigment analysis have to be revised. Firstly, I cannot understand whether LC-MS or HPLC-MS was used? Did the Authors use carotenoids standards? If Yes, they should be indicated. As a rule, the pigments are eluated based on their polarity, and astaxanthin esters goes before β-carotene, however another is seen of the chromatogram. Is it possible to show representative m/z values for protonated carotenoid esters? As I can see from ref [37], the authors used TOF ionization mode. Isn't it? If Yes, the fragmented ions were detected and have to be involved into analysis. Is it also possible to compare the spectra with published data (eg Rivera et al. (2014) Mass Spectrometry Reviews, 33(5), 353-372).

-The authors should explain in more detail calculation of growth parameters, e.g. specific growth rates. Based on which parameter (optical density)? It is imortant also to compare them with published data for Coelastrella. I exists at least for Na acetate.

-l. 171-185. It is also interesting to indicate the fraction of carotenoids in algal biomass and compare it with published data.

-Strain of the alga should be indicated through the text (it is not only the name of the species).

-l. 205. It is better to have a reference to original descriptions (Hegewald, Hanagata. (2002). Algological Studies/Archiv für Hydrobiologie, Supplement Volumes 105: 7-9 or the description by Reisigl).

-l. 231. What is PCR cultivation? PBR?

-l. 235-237. This conclusion should be revised. Why osmotic stress? Why not pH. Did You fix pH parameters? Relatively low acetate concentration was used (12 mM), it should not increase significantly osmotic pressure.

-l. 334-335. I think this experimental strategy was not correct. Molar mass of different C sources in the work. It is more correct to use the same molar concentrations or concentrations recalculated to the number of C atoms in molecules. It is the limitation of the study.

-l. 51. Actually, "cysts" of H. lacustris are not cysts. They are aplanospores.

-l. 82. GenBank accession number should be provided at first mention.

-Haematococcus pluvialis is a wrong obsolete synonym. Haematococcus lacustris is correct. It was considered that H. pluvialis is the invalid synonym due to priority of the name H. lacustris (please, see Nakada, T., & Ota, S. (2016). What is the correct name for the type of Haematococcus Flot.(Volvocales, Chlorophyceae)?. Taxon, 65(2), 343-348). It is also mentioned in Algaebase. Moreover the name H. pluvialis from 2017 banned by NCBI GenBank.

Author Response

It has been decided to revise the article. More detailed analysis of presented data is necessary. Moreover, I cannot agree with some responses of the Authors.

-Although the Authors tried to explain the novelty of the work and cited some recent papers, the novelty is weak. (1) previously I provided some work on carotenoid composition in Coelastrella spp., reporting one more strain has only a local significance, (2) probably, the novelty could be related to use a system including the strain and photobioreactor; in this case they should provide its construction as a scheme or figure, not only verbal description, (3) adonixanthin accumulation is not novel for microalgae, particularly in significant amounts (see e. g. the work Yuan et al. (2002). Food Chemistry, 76(3), 319-325). Moreover, the Authors did mention even their own recently published paper (Doppler et al. (2021)., first reported member of hydrodictyaceae to accumulate secondary carotenoids. Life, 11(2), 107.) on Tetrahedron minutum, in which the fraction of adonixanthin was also significant. To highlight the novelty, they should at least compare obtained data with this publication. What was the difference (except one more strain of algae)? Unfortunately, the Authors cited this work in the methodological section only. Another possible novelty is the growth on different carbon sources, but they did not focus on it as could be seen from the title and abstract.

We would like to respond to these three comments provided by Reviewer 1 as follows:

  • We want to point out that Reviewer 1 refers to two manuscripts not dealing with Coelastrella and three manuscripts concerning Coelastrella, of which, however, only the work of Minyuk et al. (2016) is relevant for our study. We were aware of this paper, however we did not refer to it in our manuscript since the authors did not detect or mention any adonixanthin in the work. The same group (Minyuk et al. 2017) published another study where they showed adonixanthin accumulation of their C. rubescens strain to which we are referring to as the only report of adonixanthin production with Coelastrella in our Introduction section.

The other two papers dealing with Coelastrella spp., Reviewer 1 mentions, were: Kawasaki et al. (2019), who did not identify a single carotenoid but showed phylogenetic analysis and microscopic images of this species; and Zaytseva et al. (2021) who also did phylogenetic analysis as well as carotenoid identification after separation via thin layer chromatography, subsequent extraction and absorption spectra measurement. However, this study is rather distant to the work we are presenting here, which we are not referring to it in our manuscript.

(2) As requested, we provided a picture of our stirred photobioreactor system in the Supplementary Material file. We also improved our introduction to explained in detail, in which aspects we are differentiating from other publication involving Coelastrella spp. We hope that our motivation has become clear now.

(3) We agree, adonixanthin accumulation is not novel to green algae as shown by Yuan et al. (2002, doi: 10.1016/S0308-8146(01)00279-5) for Chlorococcum sp. However, the authors did not measure absolute carotenoid content and did not test any biotechnological use of their strain. In our previously published manuscript, Doppler et al. 2021, we did identify adonixanthin. However, this paper was on the first verification of secondary carotenoids in the family Hydrodictyaceae. As indicated in that publication, this strain is poorly growing, very sensitive and therefore not suitable as potential production strain for adonixanthin. Nevertheless, we compared the accumulated amounts of adonixanthin in the revised Discussion section.

-First of all, I cannot understand the general organization of data. Microscopic photographs of the strain are located in the main body of the text, whereas more precise identification by phylogenetic analysis is given as supplementary. The same is true about the data of carotenoid identification by HPLC-MS (or LC-MS, I cannot understand from the text). Since carotenoid composition is discussed, this data plays a key role in the work, but chromatogram and results of MS are also given only as supplementary data. Moreover, these two parts of the work have to be substantially revised.

The authors appreciate this comment. We reorganized the subsection describing the identification of pigments. Therefore, the chromatogram of HPLC separation and the results of MS analysis have been added to the main manuscript.

-Phylogenetic analysis: to determine the strain at the level of species, the phylogenetic tree should not include the sequences mentioned as Coelastrella sp., particularly closely related to the studied strain. The tree must contain authentic strains, these strains have to be indicated. CCALA 476 is not authentic. If it will not be done, the strain should be defined as Coelastrella sp.

We thank for this suggestion. We recalculated the tree by the online tool in the publicly accessible NCBI database to show the distances of related species. In the revised tree, we highlighted the authentic/type strain.

-The procedure of phylogenetic analysis should be revised. "It was calculated by the Maximum Likelihood method by 1000 bootstrap replications" - the ML tree cannot be calculated by bootstrap procedure. It is built based on the calculation and maximisation of the LogLikelihood function for a selected DNA evolution model, whereas bootstrap is the procedure for testing the robustness of tree topology. The reference for the ML algorithm, bootstrap and selected model are required. What was the algorithm of multiple alignment, what was the criterion for model selection, how was the initial tree for ML analysis selected, what was the heuristic search method?

Obviously, Reviewer 1 is specialist in the area of phylogeny. Following the suggestions, we recalculated the phylogenetic tree by the NCBI database tool and highlighted type strains. We show the tree as supplementary material, since our study is not dealing with phylogeny, but with the biotechnological production of adonixanthin (as also stated in the title of our study). For further analysis the Gene bank accession number is noted in the manuscript.

-Results of pigment analysis have to be revised. Firstly, I cannot understand whether LC-MS or HPLC-MS was used? Did the Authors use carotenoids standards? If Yes, they should be indicated. As a rule, the pigments are eluated based on their polarity, and astaxanthin esters goes before β-carotene, however another is seen of the chromatogram. Is it possible to show representative m/z values for protonated carotenoid esters? As I can see from ref [37], the authors used TOF ionization mode. Isn't it? If Yes, the fragmented ions were detected and have to be involved into analysis. Is it also possible to compare the spectra with published data (eg Rivera et al. (2014) Mass Spectrometry Reviews, 33(5), 353-372).

The authors thank the Reviewer for bringing up these questions. However, we do not agree with some criticisms and want to provide some information on carotenoid analysis:

  1. i) The separation technology is called “liquid chromatography (LC)” and in combination with mass spectrometry (MS) it is called “LC-MS”. HPLC was formerly known as high pressure LC, which indicated the use of a pump for generating pressure for higher flow rates of the mobile phase through the column compared to the relatively low pressure caused by gravity forces maintaining the flow in “normal LC”. Nowadays, it is known as high performance liquid chromatography, but “high performance” is not defined – it was invented for marketing purposes. Therefore, LC-MS and HPLC-MS are basically the same and the term LC-MS is conventionally used in literature.
  2. ii) We used astaxanthin standards. We added the respective information in the Materials & Methods section in the revised manuscript.

iii) As visible in Fig. 2 showing the HPLC separation, astaxanthin and adonixanthin monoesters elute before β-carotene, as expected. However, when astaxanthin is esterified on both ionone rings by a fatty acid (astaxanthin diester), the whole molecule becomes less polar (due to alkyl backbone of FA) and thus, travels more slowly through the resin. The order of elution is: adonixanthin and astaxanthin monoesters – b-carotene – astaxanthin diesters. Exactly the same order is shown in several other publications, e.g., also by Grewe et al. 2008 (10.1002/biot.200800067) for H. pluvialis.

  1. iv) It would be possible to show theoretical m/z for all possible combinations of adonixanthin monoesters and astaxanthin mono- and diesters. Considering all possible fatty acids from C:12 to C:24 in all degrees of unsaturation would yield in hundreds of theoretical masses. Therefore, these values are not shown here.
  2. v) Time of flight (TOF) is not an ionization mode. TOF is used in mass spectrometry to calculate the masses of the ions. Therefore, TOF is the detection method. Atmospheric pressure chemical ionization (ACPI) in positive mode was used for our evaluations, as we explained in the revised Materials & Methods section.
  3. vi) We identified the (M+H)+ ion of all carotenoids shown in the Results section. Additionally, we confirmed all pigments by their characteristic light absorption spectra. In the community this is a widely accepted procedure. Clearly, also fragment ions could be evaluated to underline the findings. However, for this, reference libraries would be needed for automated spectra comparison and are currently not available.

-The authors should explain in more detail calculation of growth parameters, e.g. specific growth rates. Based on which parameter (optical density)? It is imortant also to compare them with published data for Coelastrella. I exists at least for Na acetate.

We added specific information and equations on how we calculated specific growth rates and other rates in the revised Materials & Methods section.

-l. 171-185. It is also interesting to indicate the fraction of carotenoids in algal biomass and compare it with published data.

The information of the carotenoid fraction of dried biomass was already shown in the Supplementary Material.

-Strain of the alga should be indicated through the text (it is not only the name of the species).

Revised as requested.

-l. 205. It is better to have a reference to original descriptions (Hegewald, Hanagata. (2002). Algological Studies/Archiv für Hydrobiologie, Supplement Volumes 105: 7-9 or the description by Reisigl).

Revised as requested.

-l. 231. What is PCR cultivation? PBR?

Revised as requested.

-l. 235-237. This conclusion should be revised. Why osmotic stress? Why not pH. Did You fix pH parameters? Relatively low acetate concentration was used (12 mM), it should not increase significantly osmotic pressure.

Green algae are sensitive to salt and even low amounts may cause reduced biomass growth. These effects on H. pluvialis were recently extensively reviewed by Xi et al. (2020, 10.1016/j.biotechadv.2020.107602). 12 mM NaOAc is 1 g/L which is almost the double amount of total salt content of the BBM medium. As stated in our Materials & Methods section we used automated pH control. Hence, we could exclude severe pH drops as in uncontrolled Erlenmeyer flask experiments of Minyuk et al. (2017) who hypothesized pH drops were the main stress factor causing carotenoid accumulation in C. rubescens.

-l. 334-335. I think this experimental strategy was not correct. Molar mass of different C sources in the work. It is more correct to use the same molar concentrations or concentrations recalculated to the number of C atoms in molecules. It is the limitation of the study.

Pre-experiments for initial substrate screening were done by the addition of 1.0 g/L of different C-sources. By conducting these experiments, we just wanted to find out, which C-sources are metabolized at all by our strain. The outcomes of these experiment (fastest initial growth) were calculated based on OD600 values within the first 10 days of cultivation. Until then, all cultures still contained organic carbon and nitrogen, as explained in the manuscript. For PBR cultivations we provided equimolar amounts of C-atoms as indicated in the Materials & Methods section.

-l. 51. Actually, "cysts" of H. lacustris are not cysts. They are aplanospores.

Revised as requested.

-l. 82. GenBank accession number should be provided at first mention.

We provided the accession number in the section Materials & Methods subsection 4.2. Strain Identification.

-Haematococcus pluvialis is a wrong obsolete synonym. Haematococcus lacustris is correct. It was considered that H. pluvialis is the invalid synonym due to priority of the name H. lacustris (please, see Nakada, T., & Ota, S. (2016). What is the correct name for the type of Haematococcus Flot.(Volvocales, Chlorophyceae)?. Taxon, 65(2), 343-348). It is also mentioned in Algaebase. Moreover the name H. pluvialis from 2017 banned by NCBI GenBank.

Revised as requested.

Reviewer 3 Report

Although I requested for my review to be edited because of a copy and paste error, the authors did not improve Fig. 4. I request again for the authors to improve these figures as they are incorrect. The original text was as follows:

  • Fig. 4 is quite odd. The legend refers to two panels and only one panel is shown. Moreover, the inset legend refers to BBM and 2N-BBM but no data points are visible for these media (there are no black data points as in the inset legend). I think a panel is missing because the text refers to Figure 4a and 4b. Please correct this.

Author Response

Although I requested for my review to be edited because of a copy and paste error, the authors did not improve Fig. 4. I request again for the authors to improve these figures as they are incorrect. The original text was as follows:

Fig. 4 is quite odd. The legend refers to two panels and only one panel is shown. Moreover, the inset legend refers to BBM and 2N-BBM but no data points are visible for these media (there are no black data points as in the inset legend). I think a panel is missing because the text refers to Figure 4a and 4b. Please correct this.

The authors thank for bringing up this question. We have received this Reviewer’s request before. Thus,  we removed the reference to two different panels, since only one panel/figure is shown. Indeed, the inset legend referred to BBM and 2N-BBM and these data points are distinguished by two different symbols. For an easier interpretation we removed the “media legend” and described the two symbols for BBM and 2N-BBM in the figure legend. We hope the Reviewer is fine with this solution!

Round 3

Reviewer 1 Report

The authors have tried to correct most of my concerns.  Although I am not completely satisfied, I will not block any longer acceptance.

For instance:

-I completely disagree with the response on the strain identification: "We show the tree as supplementary material, since our study is not dealing with phylogeny" - if the Authors do not feel that thay cannot give a correct analysis in the frame of one work, they should published a separate work on identification, e.g. in a more specialized journal, and should not say that they performed identification. In any way, If the data are presented, it must be presented only in accordance with current standards. Otherwise it is better not give the data at all.

-Unfortunately, they did not compare the data with previous work in Tetraedron. However it plays a key role in the statement of novelty. In previous work it was stated that “Finally, insights were given about the stress mechanisms and the kinetics of secondary carotenoid formation, and future methodological optimizations have to aim for an increase in the yield”, moreover, it was grown in photobioreactors. Therefore it was considered as a possible source of adonixanthin in this microalgae. They provided an explanation why they did not consider this strain (Tetraedron) in the introduction but did not say it in the text. Thus, in the current state, it makes an  erroneous feeling that this data is not discussed in detail to artificially increase the novelty of current work on Coelastrella

-The response on the m/z ratio of carotenoid esters: tables must not have empty cells.

It remains to be hoped that the Authors will publish further works  more consistent with the high quality of the Journal.

Author Response

“The authors have tried to correct most of my concerns. Although I am not completely satisfied, I will not block any longer acceptance.
For instance:
-I completely disagree with the response on the strain identification:
"We show the tree as supplementary material, since our study is not dealing with phylogeny" - if the Authors do not feel that thay cannot give a correct analysis in the frame of one work, they should published a separate work on identification, e.g. in a more specialized journal, and should not say that they performed identification. In any way, If the data are presented, it must be presented only in accordance with current standards. Otherwise it is better not give the data at all.

Our study deals with the biotechnological production of adonixanthin with a microalga. After the first round of review, both Reviewer 1 and 3 suggested to calculate a phylogenetic tree for strain identification. Reviewer 3 even gave rather detailed instructions on how exactly he/she would calculate the tree and we followed these suggestions precisely. Consequently, Reviewer 3 accepted the phylogenetic tree, whereas Reviewer 1 still had some issues, however did not give suggestions for improvement. Still, we calculated another phylogenetic tree by the tool of NCBI with highlighted authentic strains, as requested by Reviewer 1. There, one can clearly see the clustering of our strain to other Coelastrella terrestris strains. We left the tree in the Supplementary Material, since we only used phylogeny for identification. We never hypothesized about a new species or any relationship to other species of genus Coelastrella.

-Unfortunately, they did not compare the data with previous work in Tetraedron. However it plays a key role in the statement of novelty. In previous work it was stated that “Finally, insights were given about the stress mechanisms and the kinetics of secondary carotenoid formation, and future methodological optimizations have to aim for an increase in the yield”, moreover, it was grown in photobioreactors. Therefore it was considered as a possible source of adonixanthin in this microalgae. They provided an explanation why they did not consider this strain (Tetraedron) in the introduction but did not say it in the text. Thus, in the current state, it makes an erroneous feeling that this data is not discussed in detail to artificially increase the novelty of current work on Coelastrella.

In the Discussion section we in fact compared the data with our previous work on Tetraedron minimum in terms of adonixanthin productivity, as requested by Reviewer 1. So, we cannot follow this criticism. However, the novelty of the current study, as stated several times in the manuscript, is the evaluation and quantification of secondary carotenoid (and in special adonixanthin) productivities of Coelastrella in photobioreactor cultivations. We are not aware of such studies in literature.

-The response on the m/z ratio of carotenoid esters: tables must not have empty cells.

The m/z values of carotenoid esters are not shown, since this would yield in hundreds of possible values, as mentioned in our last revision letter. So again, we cannot follow this statement.

It remains to be hoped that the Authors will publish further works more consistent with the high quality of the Journal.

We do not dignify this rather hostile and unnecessary comment with a response.

We sincerely hope that you share our enthusiasm for this work and our vision of its potential broad interest and impact, and for positive feedback from your side.